# How the Choice of Distance Measure Influences the Detection of Prior-Data Conflict

**DOI:** 10.3390/e21050446

**Published:** 2019-04-29

**Authors:** Kimberley Lek, Rens Van De Schoot

**Affiliations:** 1Department of Methods and Statistics, Utrecht University, 3584 CH 14 Utrecht, The Netherlands; 2Optentia Research Program, Faculty of Humanities, North-West University, Vanderbijlpark 1900, South Africa

**Keywords:** prior-data conflict, distance measure, Kullback-Leibler, data agreement criterion

## Abstract

The present paper contrasts two related criteria for the evaluation of prior-data conflict: the Data Agreement Criterion (DAC; Bousquet, 2008) and the criterion of Nott et al. (2016). One aspect that these criteria have in common is that they depend on a distance measure, of which dozens are available, but so far, only the Kullback-Leibler has been used. We describe and compare both criteria to determine whether a different choice of distance measure might impact the results. By means of a simulation study, we investigate how the choice of a specific distance measure influences the detection of prior-data conflict. The DAC seems more susceptible to the choice of distance measure, while the criterion of Nott et al. seems to lead to reasonably comparable conclusions of prior-data conflict, regardless of the distance measure choice. We conclude with some practical suggestions for the user of the DAC and the criterion of Nott et al.

## 1. Introduction

Any Bayesian model consists of at least two ingredients: a prior and a sampling model for the observed data. The prior contains the a priori beliefs about the parameter(s) and can, for instance, be based on expert knowledge, if elicitation methods are used [1,2,3,4,5,6]. Consecutive inferences based on the Bayesian model are built on the assumption that the (expert) prior is appropriate for the collected data. In case of a prior data conflict, this assumption is violated. In such a conflict, the prior primarily favors regions of the parameter space that are far from the data mass. Such a conflict can be caused by the unfortunate collection of a rare data set, a problem with the prior, or both of these factors [7]. To avoid erroneous inferences, checking for prior-data conflict should be ‘part of good statistical practice’ ([8], p. 894).

In the current paper, we focus on two checks, developed for the detection of prior-data conflict. First, Bousquet (see [8]; see also [9]) suggested that the expert prior be compared with a non-informative, reference prior in his “Data Agreement Criterion” (DAC). When the expert prior has a larger distance to a reference posterior—a posterior based on the data and the non-informative, reference prior—than the non-informative prior, it can be concluded that the expert and data are in conflict. Second, Nott et al. (see [10]) suggested that the distance between the expert prior and resulting posterior be measured directly. According to this method, the expert and data are in conflict when this distance is surprising in relation to the expert’s prior predictive distribution (see also [11]).

One thing that the DAC [7] and the criterion of Nott et al. [10] have in common is that they depend on a distance measure. Currently, both the DAC and the Nott et al. criteria rely on the Kullback-Leibler divergence (heretofore abbreviated as KL). One of the advantages of the KL is that it has an intuitive interpretation—the informative regret due to the prior choice—and some favorable analytical properties, such as its invariance to reparameterization [12]. However, the KL is not the only available option for the measurement of distance between statistical distributions. To illustrate: the “Encyclopedia of distances” [13] consists of 583 pages filled with distance measures, and their list is not even exhaustive. How the DAC and the Nott et al. criteria behave for different distance measure choices has yet to be investigated.

Therefore, the goal of the current paper is to investigate how the implementation of different distance measures in the DAC and Nott et al. criteria influence the detection of prior-data conflict. In the remainder, we first discuss both prior-data conflict criteria in detail. Thereafter, we discuss the design and results of a simulation study into the effect of the choice of distances on the conclusion of prior-data conflict, using a variety of distance measures (also see the Appendix A for an overview). We end with a general discussion and practical recommendations for users, who would like to test for a prior-data conflict.

## 2. Prior-Data Conflict Criteria

### 2.1. DAC

#### 2.1.1. Computation of Prior-Data Conflict

The Data Agreement Criterion (DAC; [7]) is based on the ratio of two distances, denoted by ∆. The first distance is between the expert prior $\pi_{i}^{E}\left(\theta \right)$ and the posterior $\pi^{B}\left(\theta |y_{n}\right)$, where θ denotes the parameter of interest, and upper script *B* is used for a ‘benchmark’, upper script *E* for an ‘expert’, and subscript *i*, 1, ⋯, *I* to distinguish between experts. As reflected in the expression, ‘$\pi^{B}\left(\theta |y_{n}\right)$’, the posterior is based on dataset $y_{n}$ and prior $\pi^{B}\left(\theta \right)$. The idea is to choose $\pi^{B}\left(\theta \right)$, such that the posterior is dominated by the data $y_{n}$. Following [14,15], in order to affect the posterior as little as possible, a reference prior should be chosen for $\pi^{B}\left(\theta \right)$. The second distance is between the posterior $\pi^{B}\left(\theta |y_{n}\right)$ and the non-informative prior $\pi^{B}\left(\theta \right)$. The ratio between the two distances results in the DAC:(1)$${\mathrm{DAC}}_{i}=\;\frac{∆\left(\pi^{B}\left(\theta |y_{n}\right)||\pi_{i}^{E}\left(\theta \right)\right)}{∆\left(\pi^{B}\left(\theta |y_{n}\right)||\pi^{B}\left(\theta \right)\right)}.$$

In [7,9], the KL-divergence is used for ∆, denoted here by ${∆}_{KL}$. The KL-divergence expresses the loss of information that occurs when we rely on the expert prior $\pi_{i}^{E}\left(\theta \right)$, instead of on the posterior $\pi^{B}\left(\theta |y_{n}\right)$:(2)$${∆}_{KL}\left(\pi^{B}\left(\theta |y_{n})\right\|\pi_{i}^{E}\left(\theta \right)\right)=\;\int_{\Theta }\pi^{B}\left(\theta |y_{n}\right)\;\log\frac{\pi^{B}\left(\theta |y_{n}\right)}{\pi_{i}^{E}\left(\theta \right)}\;d\theta .$$

Here, Θ is the set of all possible values for the parameter θ. Figure 1a illustrates the KL-divergence between the prior $\pi^{B}\left(\theta \right)$ and posterior $\pi^{B}\left(\theta |y_{n}\right)$. The lower part of this figure shows the prior $\pi^{B}\left(\theta \right)$ (dashed line) and the $\pi^{B}\left(\theta |y_{n}\right)$ (solid line), and the upper part of this figure shows the corresponding KL-divergence, which is equal to the highlighted area under the curve. Figure 1b illustrates the KL-divergence between the expert prior $\pi_{i}^{E}\left(\theta \right)$ and the posterior $\pi^{B}\left(\theta |y_{n}\right)$ (see [9]). To compute the ${\mathrm{DAC}}_{i}$ the KL-divergence in the upper part of Figure 1b is divided by the KL-divergence in the upper part of Figure 1a. In this example, the KL-divergence for the prior $\pi^{B}\left(\theta \right)$ is 1.26, and the KL-divergence for the expert prior $\pi_{i}^{E}\left(\theta \right)$ equals 0.89. Since this value is lower than 1.26 (the “benchmark” KL-divergence), we would, in this example, conclude that there is no prior-data conflict, according to the DAC criterion.

Note that the DAC (Equation (1) does not necessarily have to be based on the KL-divergence (Equation (2); any other distance or divergence measure can be substituted for ∆.

#### 2.1.2. Definition of Prior-Data Conflict

As is shown by Equation (1), the distance between the prior $\pi^{B}\left(\theta \right)$ and posterior $\pi^{B}\left(\theta |y_{n}\right)$ serves as a benchmark. When the distance between the expert prior $\pi_{i}^{E}\left(\theta \right)$ and posterior $\pi^{B}\left(\theta |y_{n}\right)$ exceeds the benchmark (i.e., when ${\mathrm{DAC}}_{i}>1$), it is concluded that there is a prior-data conflict. As stated by Bousquet (2008), $\pi^{B}\left(\theta \right)$ can be seen as a fictitious, oblivious expert, perfectly in agreement with $y_{n}$. When this ‘ideal’, fictitious expert $\pi^{B}\left(\theta \right)$ is *more* in agreement with $\pi^{B}\left(\theta |y_{n}\right)$ than the actual expert, it is concluded that there is a prior-data conflict. According to Bousquet [7], this happens in the following two scenarios: (1) when the expert ($\pi_{i}^{E}\left(\theta \right))$ favors the regions of Θ that are far from the data mass (conflict in location), or (2) when the prior information on θ is far more precise than the information from the data $y_{n}$ (conflict in information uncertainty). The latter scenario mainly arises with small data sets. Note that a prior-data conflict, as defined by Bousquet [7], does not necessarily point to a problem with the expert prior. The DAC is developed as a simultaneous check of the sampling model, and the expert data and may both have symmetric roles in the conflict.

#### 2.1.3. Pros and Cons

The DAC is appealing to the applied user because of its clear, binary decision. Other than with *p*-values (see [10]; next section), the applied user does not have to choose a threshold him- or herself. A disadvantage, however, is the necessity to specify a suitable non-informative, benchmark prior $\pi^{B}\left(\theta \right)$. Choosing such a prior is a delicate matter, since there are many alternative candidates to be considered, which all directly influence the conclusion and thus the definition of the prior-data conflict. This is especially problematic when an improper prior is chosen. In that case, the notion of distance gets lost in the determination of $∆\left(\pi^{B}\left(\theta |y_{n}\right)||\pi^{B}\left(\theta \right)\right)$. It might also feel counterintuitive that, in order to compare an expert prior $\pi_{i}^{E}\left(\theta \right)$ with $y_{n}$, the specification of another prior $\pi^{B}\left(\theta \right)$ is required.

### 2.2. Criterion of Nott et al.

#### 2.2.1. Computation of Prior-Data Conflict

Inspired by the work of Box see [16] and Evans and Moshonov [8,17,18], the criterion developed by Nott et al. [10] is based on the premise of Bayesian prior predictive model checking. The prior predictive distribution informs us as to which data are considered plausible a priori by the respective expert. When the collected data $y_{n}$ are surprising under the prior predictive (i.e., located in the tails of the prior predictive distribution), this signals that something is wrong. More formally stated, the idea is that there is a discrepancy function D($y_{n})$ of the data $y_{n}$ (see [19]) and that, for the prior predictive distribution, a Bayesian *p*-value is computed as (see p. 3, [11]):(3)$$p=P\left(D\left(Y\right)\geq D\left(y_{n}\right)\right),$$
where *Y* is a draw from the prior predictive distribution. A small *p*-value results when *D*($y_{n})$ is large (i.e., surprising), compared to D(Y). In principle, any discrepancy function can be chosen that suits a specific model-checking goal. To check for prior-data conflict, Evans and Moshonov [8] consider $1/m_{T}\left(T\left(y_{n}\right)\right)$ as their discrepancy function, where $m_{T}$ is the prior predictive density of a minimal sufficient statistic *T*. Despite their promising results (see [17,18]), a drawback of this suggestion is that the check is not invariant in relation to the choice of the minimal sufficient statistic, when (3) is continuous (see [20]). Nott et al. [10] solve this problem elegantly by considering another discrepancy function: the distance between the prior $\pi_{i}^{E}\left(\theta \right)$ and posterior $\pi_{i}^{E}\left(\theta |y_{n}\right)$; $∆\left(\pi_{i}^{E}\left(\theta |y_{n}\right)||\pi_{i}^{E}\left(\theta \right)\right)$. The advantage of $∆\left(\pi_{i}^{E}\left(\theta |y_{n}\right)||\pi_{i}^{E}\left(\theta \right)\right)$ is that it depends on the data $y_{n}$
*only* through the posterior distribution $\pi_{i}^{E}\left(\theta |y_{n}\right)$. Hence, the statistic $∆\left(\pi_{i}^{E}\left(\theta |y_{n}\right)||\pi_{i}^{E}\left(\theta \right)\right)$ is a function of any sufficient statistic and thus invariant in relation to the particular choice of *T*.

To express the distance between $\pi_{i}^{E}\left(\theta \right)$ and $\pi_{i}^{E}\left(\theta |y_{n}\right)$, Nott et al. [10] consider a class of divergences, of which the KL-divergence is a special case. This class contains the Rényi divergences of order α, here denoted by ${∆}_{R\alpha }$. ${∆}_{R\alpha }$ can be explained as a measure of how much beliefs change from prior to posterior, comparable to the notion of relative belief (see [21]). The earlier mentioned KL-divergence is a special case of this class, with α→1. For the Nott et al. [10] criterion, Equation (3) can thus be rewritten as:(4)$$p_{i}=P\left({∆}_{R\alpha }\left(\pi_{i}^{E}\left(\theta |Y\right)\pi_{i}^{E}\left(\theta \right)\right)\geq {∆}_{R\alpha }\left(\pi_{i}^{E}\left(\theta |y_{n})\right\|\pi_{i}^{E}\left(\theta \right)\right)\right),$$
where
(5a)$${∆}_{R\alpha }\left(\pi_{i}^{E}\left(\theta |y_{n})\right\|\pi_{i}^{E}\left(\theta \right)\right)=\;\frac{1}{\alpha -1}\;\log\int_{\Theta }\pi_{i}^{E}\left(\theta |y_{n}\right){\left\{\frac{\pi_{i}^{E}\left(\theta |y_{n}\right)}{\pi_{i}^{E}\left(\theta \right)}\right\}}^{\alpha -1}d\theta ,$$
and
(5b)$${∆}_{R\alpha }\left(\pi_{i}^{E}\left(\theta |Y)\right\|\pi_{i}^{E}\left(\theta \right)\right)=\;\frac{1}{\alpha -1}\;\log\int_{\Theta }\pi_{i}^{E}\left(\theta |Y\right){\left\{\frac{\pi_{i}^{E}\left(\theta |Y\right)}{\pi_{i}^{E}\left(\theta \right)}\right\}}^{\alpha -1}d\theta .$$

Figure 2 illustrates the criterion of Nott et al. for the same $\pi_{i}^{E}\left(\theta \right)$ and $y_{n}$, as used in Figure 1. In Figure 2a, the KL-divergence between the prior $\pi_{i}^{E}\left(\theta \right)$ (lower panel; dashed line) and posterior $\pi_{i}^{E}\left(\theta |y_{n}\right)$ (lower panel; solid line) is illustrated. The area under the curve is equal to 0.43. In Figure 2b, this value (0.43) is compared to 10^5^ KL-divergences, obtained by repeatedly drawing Y from the prior predictive distribution of the expert *i*. In this example, 30% of the KL-divergences in Figure 2b exceed 0.43 (see the dark blue bins in Figure 2b). We would therefore conclude that there is no prior-data conflict in this example.

Note that, in prior-predictive checking, the user has considerable freedom in choosing a suitable discrepancy function D($y_{n})$. Therefore, the distance measure ${∆}_{R\alpha }$ in the criterion of Nott et al. [10] can be replaced by any other distance or divergence measure.

#### 2.2.2. Definition of Prior-Data Conflict

In [10], it is found that prior-data conflict exists when the observed distance between the prior $\pi_{i}^{E}\left(\theta \right)$ and posterior $\pi_{i}^{E}\left(\theta |y_{n}\right)$ is surprising in relation to the corresponding prior predictive distribution. In other words, if the data $y_{n}$ would have been in line with the prior $\pi_{i}^{E}\left(\theta \right)$, we would not expect to observe such a large distance ${∆}_{R\alpha }\left(\pi_{i}^{E}\left(\theta |y_{n}\right)\pi_{i}^{E}\left(\theta \right)\right)$. Hence, the prior $\pi_{i}^{E}\left(\theta \right)$ and the data $y_{n}$ appear to be in conflict.

Note that this definition of prior-data conflict is vastly different from the earlier definition of the DAC. First, although both rely on the KL-divergence, in the criterion of Nott et al., the prior $\pi_{i}^{E}\left(\theta \right)$ is directly compared to its corresponding posterior $\pi_{i}^{E}\left(\theta |y_{n}\right)$, instead of $\pi^{B}\left(\theta |y_{n}\right)$, as in the DAC. The criterion of Nott et al. (2016) and the DAC therefore ask different questions: how much information do we lose when relying on $\pi_{i}^{E}\left(\theta \right)$, instead of $\pi^{B}\left(\theta |y_{n}\right)$ (DAC)? To what extent does our information change, when moving from $\pi_{i}^{E}\left(\theta \right)$ to $\pi_{i}^{E}\left(\theta |y_{n}\right)$ (Nott et al.)? Second, the observed values ${∆}_{KL}\left(\pi^{B}\left(\theta |y_{n}\right)\pi_{i}^{E}\left(\theta \right)\right)$ and ${∆}_{R\alpha }\left(\pi_{i}^{E}\left(\theta |y_{n}\right)\pi_{i}^{E}\left(\theta \right)\right)$ are calibrated differently. In the DAC, prior-data conflict exists when more information is lost than with the benchmark prior $\pi^{B}\left(\theta \right)$. In the criterion of Nott et al., the *p*-value in Equation (4) calibrates the change in information by expressing how surprising this change is. Third, the uncertainty of the expert (i.e., the variance of $\pi_{i}^{E}\left(\theta \right)$) is treated differently in the criterion of Nott et al. than in the DAC. In the DAC, prior-data conflict can occur when there is *no* mismatch in location (i.e., both $\pi_{i}^{E}\left(\theta \right)$ and $\pi^{B}\left(\theta |y_{n}\right)$ place their mass primarily in the same region of Θ) but there *is* a mismatch in information uncertainty (i.e., the variance of $\pi_{i}^{E}\left(\theta \right)$ is smaller than the variance of $\pi^{B}\left(\theta |y_{n}\right)$). In the criterion of Nott et al., the uncertainty of the expert simply reflects the variety of data that are deemed plausible by this expert, as captured in the prior predictive distribution. Therefore, when the location of the prior and posterior match, we do not find a prior-data conflict with the criterion of Nott et al., even if that prior variance is relatively small. Finally, the DAC and the criterion of Nott et al. respond differently, when little data is available. With a small sample size, the criterion of Nott et al. may fail to detect an existing prior-data conflict. A bad prior may pass the test, as the relatively large sampling variability is taken into account in the comparison with the prior predictive. In the DAC, the sampling variability is not taken into consideration; $\pi^{B}\left(\theta |y_{n}\right)$ is treated as a temporary ‘truth’. Therefore, we should be careful not to falsely interpret a prior-data conflict as a problem with the prior in a situation where there is little data. Such a conflict may very well be the result of the unavailability of representative data and a mismatch in the amount of information captured in $\pi_{i}^{E}\left(\theta \right)$ and $\pi^{B}\left(\theta |y_{n}\right)$.

#### 2.2.3. Pros and Cons

One major advantage is that—aside from Bousquet [7]—the criterion of Nott et al. [10] does not depend on the definition of a non-informative prior $\pi^{B}\left(\theta \right)$. A possible disadvantage, however, is that the *p*-value in Equation (4) is often misinterpreted in practice (see [22]). The *p*-value in Equation (4) should be used as a measure of surprise rather than support ([20]), and not be mistaken for the probability of a (non-)conflict between $\pi_{i}^{E}\left(\theta \right)$ and $\pi_{i}^{E}\left(\theta |y_{n}\right)$. Given that the definition of prior-data conflict is quite different for the DAC and the criterion of Nott et al. (see Section 2.2.1), choosing one or the other should be based on a clear and well-informed opinion of what prior-data conflict entails.

## 3. Simulation Study

### 3.1. Goal of the Simulation

In principle, the KL-divergence in Equations (1) and (4) can be replaced by any other distance measure. The goal of our simulation is therefore to investigate the role of the distance measure in the detection of prior-data conflict. The main question is: How much does the choice of distance measure eventually impact the conclusions of prior-data conflict by the DAC and the criterion of Nott et al.? In other words: How robust are the DAC and the criterion of Nott et al. in relation to the choice of distance measure?

### 3.2. Simulation Design

#### 3.2.1. Scenario

In our simulation, we focus on one example scenario, in which θ is a one-dimensional parameter. Specifically, $y_{n}\sim$ Binomial(n, θ) and $\pi_{i}^{E}\left(\theta \right)$ is a conjugate Beta distribution beta(α,β) on [0,1]. In this scenario, $\pi^{B}\left(\theta \right)$—the benchmark prior in the DAC—is set to beta(α=1,β=1), and the prior predictive distribution for the criterion of Nott et al. equals a Beta Binomial, with parameters α,β and *n*. θ set to 0.5 in the population, whereas α,β and *n* are varied in the simulation. Note that α and β together express the location of the expert prior (i.e., the region with most of the prior’s mass), α/(α+β), and the (un)certainty of the expert. α+β. α/(α+β) are set to 0.05, 0.07, 0.09, ⋯ 0.95, respectively, and α+β is varied from 10, 12, 14, ⋯, 200. The sample size of the data, *n*, is varied between 50, 100 and 200. As explained in Section 3.2.3, twelve distance measures are compared. Figure 3 summarizes the simulation design.

#### 3.2.2. Steps

We started the simulation by taking 1000 samples from the respective population, with θ = 0.5 (see Section 3.2.1.) and sample sizes equal to 50. We repeated this sampling process for the other sample sizes (*n* = 100 and 200). Subsequently, multiple expert priors $\pi_{i}^{E}\left(\theta \right)$ were constructed, based on all possible combinations of location (i.e., α/(α+ β)= 0.05, ⋯, 0.95) and expert uncertainty (i.e., α+ β=10, 12, ⋯, 200). For all these expert priors, we then determined $\pi_{i}^{E}\left(\theta |y_{n}\right)$ (for the criterion of Nott et al.) and $\pi^{B}\left(\theta |y_{n}\right)$ (for the DAC) and calculated their respective distances to $\pi_{i}^{E}\left(\theta \right)$ using the variety of distance measures, named in Section 3.2.3. The calculation of the DAC and *p*-value of the criterion of Nott et al. (2016) followed, using Equations (1) and (4), respectively. Finally, all DAC values above 1 were flagged as prior-data conflicts. For the criterion of Nott et al. (2016), we considered *p*
≤ 0.05 as indicative of a prior-data conflict. All analyses were performed using RStudio [23] and the R packages, “Philentropy” [24], “distr” and “distrEx’ [25].

#### 3.2.3. Distance Measures

The KL-divergence, used in both the DAC and the criterion of Nott et al., is an instance of an *f*-divergence, also known as an Ali-Silvey divergence ([26]). The more general Rényi divergences, considered in Nott et al., are related to this class. One of the distance measures we investigate in our simulation (the total variation distance) belongs to the class of *f*-divergences (see [27]). We also consider distance measures that are *not* part of this class: Hellinger, Kolmogorov, Euclidean, Manhattan, Sorensen, Intersection, Harmonic mean, Bhattacharyya, Divergence, Jeffreys and Jensen-Shannon (leading to twelve distance measures in total). See [28] for more information on these distance measures.

## 4. Results

As stated in Section 3.1, the main goal of the simulation was to investigate the robustness of the DAC and the criterion of Nott et al. in relation to the choice of the twelve distance measures under study. Below, we split the results of the simulation into two parts. In the first part, we look at how the conclusion of prior-data conflict varies with different choices of distance measures, when we vary the expert priors for a specific sample $y_{n}$ (see Figure 3). Ideally, whether or not the varying expert priors are found to be in conflict with the specific sample should not depend on the choice of distance measure. In this first part, we act as if the fixed, single sample is ‘the truth’. Of course, the sample is drawn from a population. Therefore, in the second part, we look at varying samples $y_{n}$ and specific expert priors. This way, we can investigate how sampling variability influences the behavior of the DAC and the criterion of Nott et al., when different distance measures are used.

### 4.1. Robustness in Relation to the Choice of Distance Measure: Specific Sample and Varying Expert Priors

In this section, we investigate varying expert priors to determine whether they are in conflict with one specific sample, $y_{n}\sim$ Binomial(n=100, θ=0.5). To limit the number of plots here, a part of the plots is shown in the Appendix A. Specifically, the plots in the Appendix A visualize the DAC and *p*-values for varying expert priors, for each of the twelve distance measures separately. The x-axes show the expert prior uncertainty (α+ β) and the y-axis prior location (α/(α+ β)), such that, for every combination of uncertainty and location, the DAC and *p*-value can be read from the plots. The contour lines show which expert priors lead to comparable DAC and *p*-values, such as DAC =0.2, 0.3, ⋯, 1.1 and *p* = 0.05, 0.1, ⋯, 0.9. The most interesting contour lines are DAC = 1 and *p* = 0.05, since these lines distinguish the expert priors for which a prior-data conflict and no prior-data conflict is concluded. As seen in the DAC plots, the Jeffreys divergence leads to the most lenient decision of a prior-data conflict. Indeed, when Jeffreys divergence is used, relatively many expert priors lead to a DAC value below 1 (i.e., many combinations of expert uncertainty and location fall in the area of DAC values < 1). Kolmogorov, on the other hand, leads to the most stringent decision of prior-data conflict. All divergence-based measures show problematic behavior. Using ‘divergence’, DAC values below 1 are solely found for a small group of expert priors. In the criterion of Nott et al., divergence shows divergent behavior, especially for expert priors with a low α+ β. To a lesser extent, this problematic behavior at low values of α+ β is also found when using the Euclidean distance in the criterion of Nott et al. Apart from the divergence and Euclidean distance, the criterion of Nott et al. generally shows behavior more comparable to the *p*-value for different distance measures than the DAC.

Figure 4a combines the twelve DAC plots from the Appendix A into one plot, in which only the contour lines ‘DAC = 1’ are displayed. By doing so, these DAC = 1 boundaries can more easily be compared over the twelve distance measures. Figure 4b does the same for the twelve *p*-value plots from the Appendix A, for the contour lines *p* = 0.05. The contour lines (DAC = 1 and *p* = 0.05) of the distance measure which leads to the most stringent conclusion of prior-data conflict, mark the area (i.e., the combinations of expert prior uncertainty and location) for which all distance measures agree that there is *no* prior-data conflict. In Figure 4a,b, this area is colored light blue. As stated before, in Figure 4a, this area is determined by Kolmogorov. The contour lines (DAC = 1 and *p* = 0.05) belonging to the distance measure which leads to the most lenient conclusion of prior-data conflict, mark the area for which all distance measures agree that there *is* a prior-data conflict. As stated before, in Figure 4a, this area is determined by Jeffreys divergence. In Figure 4a,b, this area is colored dark blue. The area in between the contour lines of the most lenient and most stringent distance measures illustrates the area of no consensus, i.e., the expert priors for which some distance measures would lead to a conclusion of prior-data conflict and others not. The larger this area is, the more influence the choice of distance measure has on the conclusion of prior-data conflict. The white lines in this area illustrate the exact DAC = 1 and *p* = 0.05 contour lines of the distance measures. Comparing Figure 4a,b clearly shows that a different choice of distance measure more profoundly influences the conclusion of the DAC than the criterion of Nott et al. When used in the criterion of Nott et al., the distance measures primarily disagree with each other for the expert priors that have α/(α+ β) far from 0.5 (0.1, 0.2, 0.8 and 0.9) and a low α+ β (smaller than 50). The outer line separating the area with no consensus from the area with consensus on prior-data conflict is formed by the Divergence distance. Of the twelve distance measures under study here, this distance measure thus leads to the most lenient decision of prior-data conflict, at least in this specific scenario.

For only a relatively small area, the distance measures agree that there is no prior-data conflict, when used in the DAC (see Figure 4a). Other than in the criterion of Nott et al., differences in conclusions of (no) prior-data conflict are not restricted to certain combinations of prior location (α/(α+ β)) and uncertainty (α+ β).

### 4.2. Robustness in Relation to the Choice of Distance Measure: Specific Expert Prior and Varying Samples

In this section, we investigate how many times the DAC and *p*-values lead to a conclusion of prior-data conflict, when repeated samples are drawn, and the expert prior is fixed. Figure 5a–c show an overview of the results. The stacked histograms show the number of times the DAC and the criterion of Nott et al. do (red color) or do not (green color) reach a conclusion of prior-data conflict. Every stacked histogram is based on a Beta prior, with a certain combination of α/(α+ β) (row) and α+ β (column) and 1000 repeated samples $y_{n}\sim$ Binomial(n, θ) with θ = 0.5 and *n* = 50 (Figure 5a), *n* = 100 (Figure 5b) or *n* = 200 (Figure 5c). Every bin of the stacked histogram corresponds to one of the 12 distance measures. The first bin, for instance, corresponds with the total variance distance, and the second one corresponds with the Hellinger distance. When a bin within a stacked histogram is completely red, this means that *all* 1000 repeated samples led to a conclusion of prior-data conflict, when this specific distance measure was used. The dashed horizontal line and the corresponding percentage printed at the bottom of each histogram show the average number of times the distance measures lead to a conclusion of no prior-data conflict. To ease comparison, the left side of Figure 5a–c shows the results for the DAC, and the right side of Figure 5a–c shows the results for the criterion of Nott et al.

The most important conclusion that can be drawn from Figure 5a–c is that the criterion of Nott et al. is less sensitive to the choice of the twelve distance measures than the DAC. Comparing the right side (criterion Nott et al.) with the left side (DAC) of Figure 5a–c, the coloring of the bins is more comparable over the twelve distance measures when the Nott et al. criterion rather than the DAC is used. Only one of the twelve distance measures shows problematic behavior: “Divergence” (bin nr. 10). When “Divergence” is used as a distance measure in the criterion of Nott et al., the conclusion of prior-data conflict becomes more lenient. When α/(α+ β) = 0.2, α+ β = 50 and *n* = 50, for instance, almost half of the 1000 repeated samples would result in a conclusion of “no prior-data conflict”. Using any of the other distance measures would, however, lead to a conclusion of prior-data conflict in almost all samples.

## 5. Conclusions

The goal of the current paper was to investigate how the implementation of different distance measures in the DAC and Nott et al. criteria influence the detection of prior-data conflict. Overall, the criterion of Nott et al. seems less sensitive to the choice of distance measure than the DAC.

In the criterion of Nott et al., the user seems to have more freedom in the choice of distance measure. With the exception of one distance measure, all distance measures led to a comparable conclusion of prior-data conflict (see again Figure 5). This is advantageous, as even the simplest (and fastest computable) distance measures (for instance, the Manhattan distance) can be chosen, without changing the interpretation of prior-data conflict. In our specific scenario, we did see that some distance measures led to a more lenient treatment of experts, who were very insecure and incorrect.

Concerning the DAC, substituting the KL-divergence (see Equation (2)) for another distance measure is only encouraged, when the KL results in issues. One reason to *not* use the KL-divergence in the DAC, for instance, is that it can lead to computational problems (see [27]). In case of such computational problems, the best strategy is to search for a distance measure without these computational issues and with comparable behavior, so as to not jeopardize the interpretation of the DAC. Plotting the size of the DAC (or the criterion of Nott et al.), compared to the characteristics of the (possible) expert priors, as we did in Figure 4 and in the Appendix A, can help to spot potential problems with the distance measure. Such a plot can also help to check whether the distance measure leads to a relatively strict or relatively lenient conclusion of prior-data conflict.

The results of our study are limited to a specific example scenario and a limited set of twelve distance measures. We chose this limited scope, as it was impossible to investigate the hundreds of possible distance measures and anticipate all possible scenarios. Our study can, therefore, best be read as an example of how the choice of distance measure *may* influence the detection of prior-data conflict with the DAC and the criterion of Nott et al. Whether or not the detection of prior-data conflict depends on the choice of distance measure in another scenario should be checked beforehand by the investigator.

## Figures and Tables

**Figure 1 entropy-21-00446-f001:**
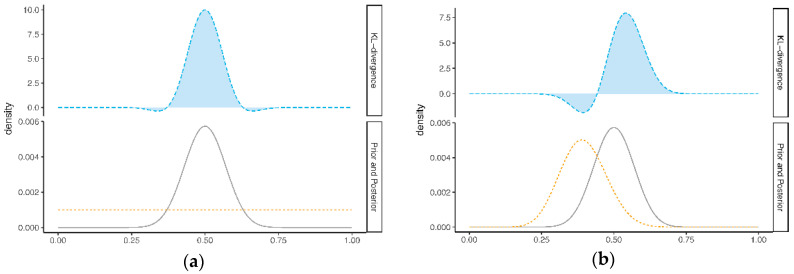
Illustration of the Data Agreement Criterion (DAC), with (**a**) the Kullback-Leibler (KL)-divergence between the posterior $\pi^{B}\left(\theta |y_{n}\right)$ and non-informative prior $\pi^{B}\left(\theta \right)$, and (**b**) the KL-divergence between the posterior $\pi^{B}\left(\theta |y_{n}\right)$ and an expert prior $\pi_{i}^{E}\left(\theta \right)$. Note the difference in the y-axis scale between the upper parts and lower parts of Figure (**a**) and (**b**).

**Figure 2 entropy-21-00446-f002:**
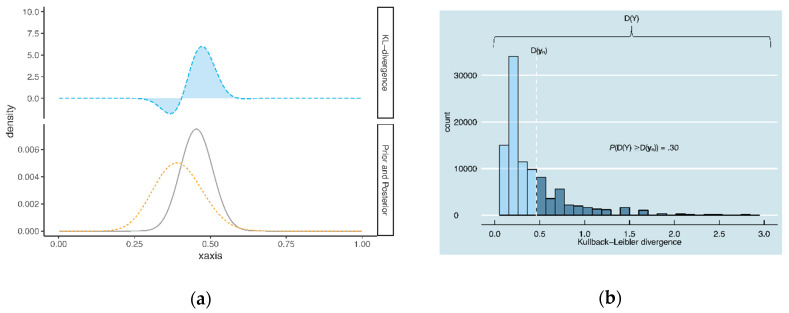
Illustration of the criterion of Nott et al. [9], with (**a**) the KL-divergence between the posterior $\pi_{i}^{E}\left(\theta |y_{n}\right)$ and expert prior $\pi_{i}^{E}\left(\theta \right)$ and (**b**) the distribution of KL-divergences between the posterior $\pi_{i}^{E}\left(\theta |Y\right)$ and expert prior $\pi_{i}^{E}\left(\theta \right)$, for 10^5^ draws Y from the prior predictive.

**Figure 3 entropy-21-00446-f003:**
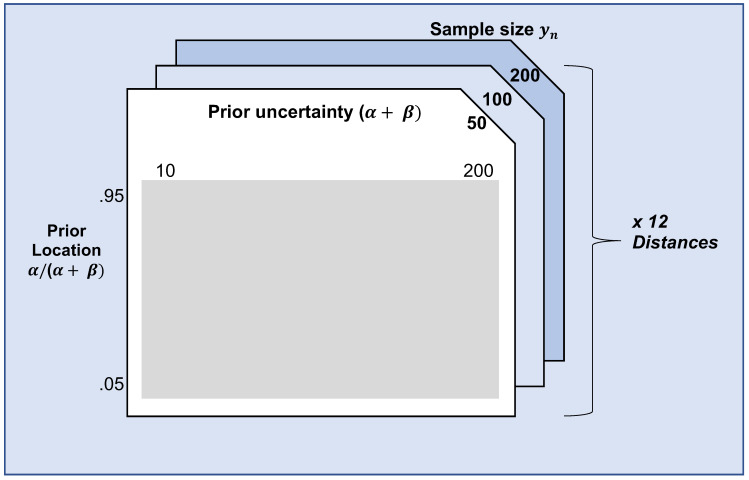
Illustration of the simulation design. The prior location (α/(α+β)), prior uncertainty (α+β), and sample size of $y_{n}$ were varied, as well as the distance measure used.

**Figure 4 entropy-21-00446-f004:**
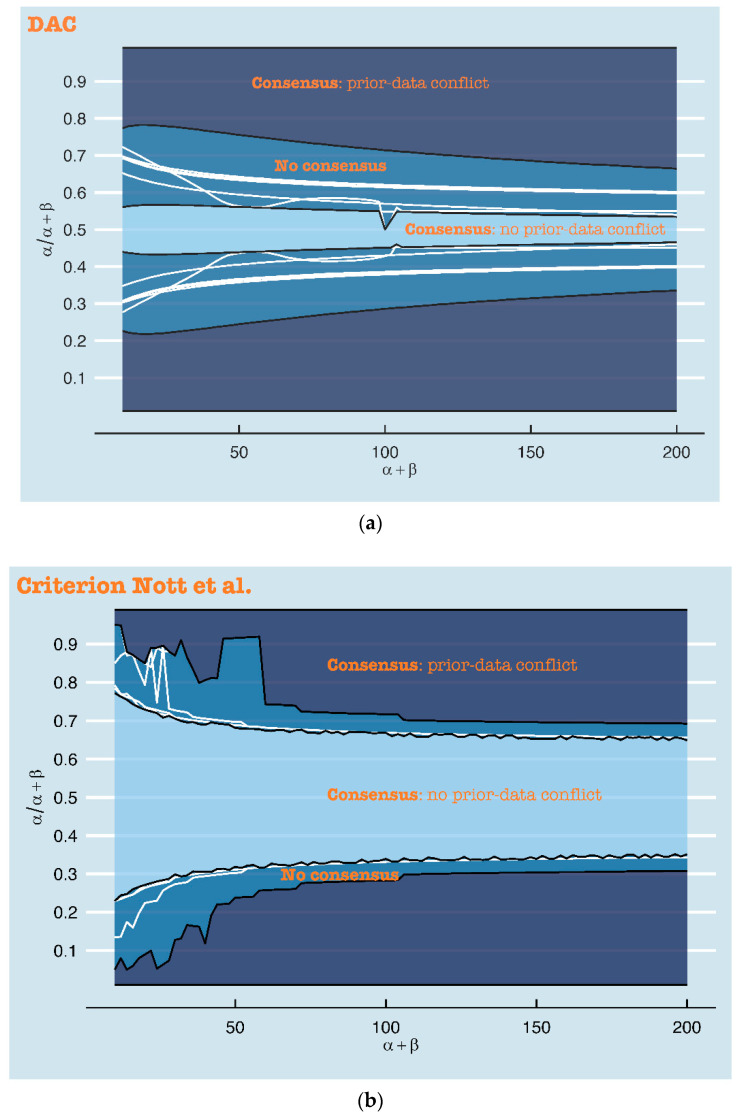
The white lines in this figure show the DAC = 1 (**a**) and *p* < 0.05 (criterion of Nott et al.) (**b**) boundaries for the twelve distance measures, when the sample is fixed, and α+ β (x-axis) and α/(α+ β) (y-axis) are varied. The darkest blue area illustrates the expert priors (i.e., the combination of α+ β and α/(α+ β)) for which all distance measures would lead to a conclusion of prior-data conflict. The lightest blue area shows the expert priors that consistently lead to a conclusion of no prior-data conflict. The area in between these two areas is inconclusive and depends on the choice of distance measure.

**Figure 5 entropy-21-00446-f005:**
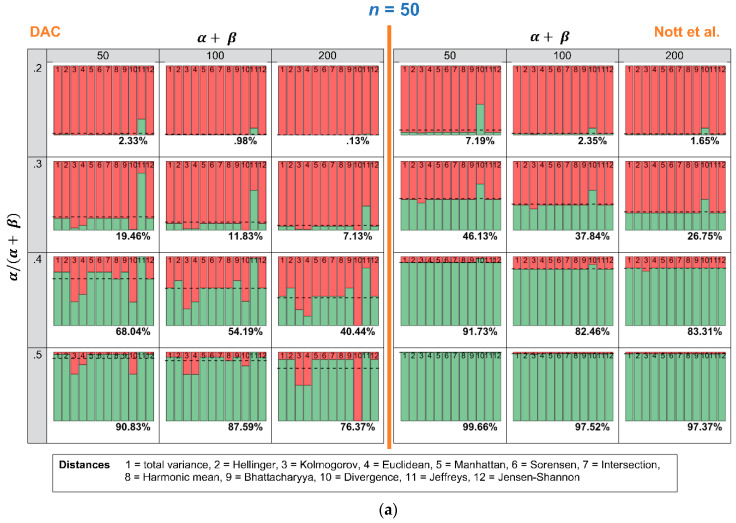
This Figure shows how many times, out of 1000 repeated samples $y_{n}\sim$ Binomial(n, θ) of θ  = 0.5 and *n* = 50 (**a**), *n* = 100 (**b**) or *n* = 200 (**c**), the twelve distance measures led to a conclusion of prior-data conflict. Every stacked histogram is based on an expert prior, with a certain combination of α/(α+ β) = 0.2, 0.3, 0.4 or 0.5 (rows) and α+ β = 50, 100 or 200 (columns). The red bins correspond to a conclusion of prior-data conflict, and the green bins correspond to a conclusion of *no* prior-data conflict.

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
