# Peer review of "How the Choice of Distance Measure Influences the Detection of Prior-Data Conflict"

_entropy, 2019, doi:10.3390/e21050446_

Round 1
Reviewer 1 Report
See attached report.

Author Response
Dear reviewer,
Thank you for your review of our article 'How the choice of distance measure influences the detection of prior-data conflict'. We have incorporated your comments in the newest version of our manuscript. Specifically:
- We have incorporated the language and small editing suggestions (i.e., comment 1, 4, 5, 6 and 9). Additionally, to be sure the article does not contain any other language deficits, we used the language editing service of Entropy.
- We added a sentence clarifying that (2) on (the previous) line 106 arises when the sample size is small (comment 2).
- We changed the discrepancy statistic in the description of the Evans and Moshonov 2006 paper (comment 3). We additionally added the word 'elegantly' when describing the Nott et al. solution, to show that the Nott et al. solution is not the only possibility (to maintain focus, we do not, however, discuss the work of Evans and Jang, 2011).
- We improved the description of Figure 4 and 5 with a more extensive description.
Thank you again for your helpful comments!
Reviewer 2 Report
This is a really extensive revision of the earlier manuscript. While the earlier version was also clearly written the new version is more clearly focused with a clearer message. The mansucript still needs a bit of editing for typographical errors and I have not tried to produce a list of these.
Author Response
Dear reviewer,
thank you for your work! To avoid any unnecessary language errors, the Entropy language editing service has checked our manuscript thoroughly.
This manuscript is a resubmission of an earlier submission. The following is a list of the peer review reports and author responses from that submission.
Round 1
Reviewer 1 Report
See attached.

Reviewer 2 Report
See the attached file
